# Physiological and Pathological Roles of the Cytohesin Family in Neurons

**DOI:** 10.3390/ijms23095087

**Published:** 2022-05-03

**Authors:** Akiko Ito, Masahiro Fukaya, Hirotsugu Okamoto, Hiroyuki Sakagami

**Affiliations:** 1Department of Anesthesiology, Kitasato University School of Medicine, Sagamihara 252-0374, Kanagawa, Japan; akikoito0810@yahoo.co.jp (A.I.); okasuke@med.kitasato-u.ac.jp (H.O.); 2Department of Anatomy, Kitasato University School of Medicine, Sagamihara 252-0374, Kanagawa, Japan; mfukaya@med.kitasato-u.ac.jp

**Keywords:** cytohesin, small GTPase, ADP ribosylation factor, neuritogenesis, chronic pain, neurodegeneration

## Abstract

The cytohesin proteins, consisting of four closely related members (cytohesins-1, -2, -3, and -4), are a subfamily of the Sec7 domain-containing guanine nucleotide exchange factors for ADP ribosylation factors (Arfs), which are critical regulators of membrane trafficking and actin cytoskeleton remodeling. Recent advances in molecular biological techniques and the development of a specific pharmacological inhibitor for cytohesins, SecinH3, have revealed the functional involvement of the cytohesin–Arf pathway in diverse neuronal functions from the formation of axons and dendrites, axonal pathfinding, and synaptic vesicle recycling, to pathophysiological processes including chronic pain and neurotoxicity induced by proteins related to neurodegenerative disorders, such as amyotrophic lateral sclerosis and Alzheimer’s disease. Here, we review the physiological and pathological roles of the cytohesin–Arf pathway in neurons and discuss the future directions of this research field.

## 1. Introduction

ADP ribosylation factors (Arfs), which were originally named after their biochemical activity as cofactors in ADP-ribosylation of the Gs subunit by cholera toxin, are small GTP-binding proteins of the Ras superfamily that play critical roles in a wide range of membrane trafficking processes in coordination with actin cytoskeleton remodeling [1,2,3,4]. In mammals, there are six Arf proteins, Arf1–6, except for the absence of Arf2 in humans [5]. They are further grouped into three classes, Arf1, Arf2, and Arf3 in class I, Arf4 and Arf5 in class II, and Arf6 in class III, based on the similarity in their amino acid sequences and genomic structures. Like other small GTPases, Arfs function as molecular on-off signaling switches by cycling between active GTP-bound and inactive GDP-bound states. Because Arfs have only slow spontaneous release rates of GDP and extremely low intrinsic GTP hydrolysis activity, the functioning of Arfs, in terms of the timing, duration, and location of their activation, depends strictly on two groups of regulatory proteins, guanine nucleotide exchange factors (GEFs) and GTPase-activating proteins (GAPs). GEFs promote GDP release from Arfs to allow for GTP binding, leading to conformational changes in the Arf proteins. These changes initiate Arf signaling by recruiting various downstream effectors and activating lipid-modifying enzymes such as phosphatidylinositol 4-phosphate 5-kinase (PIP5K) and phospholipase D (PLD). On the other hand, GAPs terminate Arf signaling by stimulating GTP hydrolysis. The GEFs (15 members) and GAPs (24 members) are more numerous than Arfs (6 members), further suggesting that the complex and fine-tuned regulation of the Arf signaling is mostly attributable to these regulatory proteins.

All Arf-GEFs share a ~200-amino acid Sec7 domain that is required for the guanine nucleotide exchange activity. The 15 Sec7 domain-containing Arf-GEFs identified to date in mammals are classified into 6 subfamilies, including the GBF1 [Golgi brefeldin A (BFA)-resistant factor 1], BIG (BFA-inhibited GEF), cytohesin, BRAG (BFA-resistant Arf-GEF)/IQSEC (IQ and Sec7 domain-containing), EFA6 (exchange factor for Arf6)/PSD [pleckstrin homology (PH) and Sec7 domain-containing], and FBX8 (F-box only protein 8) subfamilies, based on their domain organization [2,3,4,6,7].

Among these, the cytohesin subfamily comprises low-molecular-weight Arf-GEFs of ~47 kDa with unique domain organization consisting of an N-terminal coiled-coil domain, central Sec7 domain, PH domain, and C-terminal helix/polybasic region [8,9]. From the viewpoint of molecular evolution, the cytohesin subfamily is the founding member of the Arf-GEFs, along with the GBF1 and BIG subfamilies, and was already present in the last eukaryotic common ancestor [10]. Gene duplications have given rise to four mammalian cytohesin paralogs, cytohesin-1 [11], cytohesin-2 [also known as ARNO (Arf nucleotide-binding site opener)] [12], cytohesin-3 [also known as GRP1 (general receptor for phosphoinositides 1) or ARNO3] [13], and cytohesin-4 [14]. Alternative splicing further generated two isoforms of cytohesins-1, -2, and -3 with two or three glycine residues in the β1/β2 loop region of the PH domain, which differ only by the insertion of 3-bp nucleotides (GAG) encoding a glycine residue [14]. Surprisingly, the two isoforms exhibit a distinct affinity of the PH domain for phosphatidylinositol 3,4,5-trisphosphate [PI(3,4,5)P_3_]: the di-glycine variant exhibits a selective affinity for PI(3,4,5)P_3_ over phosphatidylinositol 4,5-bisphosphate [PI(4,5)P_2_], whereas the tri-glycine variant exhibits a similar affinity for PI(3,4,5)P_3_ and PI(4,5)P_2_ [15]. In fact, cytohesin-3 with the di-glycine PH domain translocates to the plasma membrane specifically upon phosphatidylinositol 3-kinase (PI3K)-dependent PI(3,4,5)P_3_ synthesis stimulated by various growth factors such as insulin, epidermal growth factor, and nerve growth factor [16,17].

The in vivo substrate specificity of the cytohesin family is still under debate. In vitro guanine nucleotide exchange assays have shown that although Arf1 is the most preferred substrate, cytohesins activate all Arf members to various extents, depending on the assay conditions [12,14,17,18,19,20,21,22]. In contrast, accumulating evidence suggests that Arf6 is a major substrate for cytohesins in various physiological contexts. For example, we have recently demonstrated that treatment with a specific pan-cytohesin inhibitor, SecinH3, blocks the activation of Arf6, but not Arf1, in the dorsal spinal cord in rodent models of inflammatory and neuropathic pain [23]. On the other hand, GTP-bound Arf6 also functions as an upstream activator of cytohesins by recruiting it to the plasma membrane and increasing its guanine nucleotide exchange activity toward Arf1 [24]. These findings raise the question of how cytohesins discriminate Arf as a substrate in cellular contexts. In addition, cytohesin-1 was also reported to bind GDP-bound Arf domain protein 1 (ARD1), a 64-kDa protein containing an 18-kDa Arf domain in its C-terminal region, and to accelerate GTP binding to ARD1 as a substrate in vitro, although the efficiency of GTP binding to Arf1 was much higher than that to ARD1 [25]. These findings raise the question of how cytohesins discriminate Arf as a substrate in cellular contexts.

In addition to these attractive features of cytohesins, the development of SecinH3 has greatly facilitated disclosure of the physiological and pathological roles of cytohesins as a whole at the cell, organ, and animal levels [26]. Indeed, accumulating evidence suggests the functional importance of the cytohesin–Arf pathway not only in normal neuronal functions, including neuronal morphogenesis [27,28,29] and synaptic transmission [30,31,32], but also in the development of chronic pain [23] and neurotoxicity induced by proteins related to neurodegenerative disorders, such as amyotrophic lateral sclerosis (ALS) [33,34] and Alzheimer’s disease (AD) [35].

In this review, we focus on the physiological and pathological roles of the cytohesin–Arf pathway in neurons and further discuss its potential as a therapeutic target for neurological disorders.

## 2. Expression and Subcellular Localization of the Cytohesin Family in the Nervous System

A previous in situ hybridization analysis revealed that three *cytohesins*, except *cytohesin-4*, are expressed in the developing and mature rat brain with distinct but overlapping spatiotemporal expression patterns [36]. Among these, *cytohesin-1* shows the most evident hybridization signals in the adult rat brain with a substantial expression level in the olfactory bulb, hippocampal formation, cerebral cortex, striatum, and cerebellar cortex. Notably, most central neurons generally express more than two *cytohesins*. For example, hippocampal pyramidal cells and dentate granule cells express *cytohesins-1*, *-2*, and *-3*, whereas Purkinje cells in the cerebellar cortex express *cytohesins-1* and *-3* at the mRNA level.

The cellular and subcellular localization of cytohesins in the central nervous system (CNS) at the protein level has so far been reported only for cytohesin-2 [23,37]. We have previously demonstrated by immunohistochemistry that cytohesin-2 is widely distributed in brain regions, particularly abundantly in the olfactory bulb, hippocampal formation, cerebral cortex, striatum, thalamus, cerebellar cortex, brain stem nuclei, and spinal cord dorsal horn [23,37]. Double immunofluorescence analyses of the hippocampal formation and spinal cord revealed that cytohesin-2 is present in various endolysosomal compartments including early, recycling, and late endosomes, and lysosomes. Further immunoelectron microscopic analyses of hippocampal neurons and spinal dorsal horn neurons revealed that cytohesin-2 is frequently associated with the plasma membrane and various intracellular membrane vesicles, including multivesicular bodies and presumed endosomes. Notably, cytohesin-2 is accumulated preferentially around the edge of the postsynaptic density in excitatory synapses, where group I metabotropic glutamate receptors (mGluRs) [23], mGluR1 and mGluR5, are also enriched [38,39,40], consistent with the finding showing the complex formation of cytohesin-2 and mGluR1/5 via a postsynaptic scaffold protein, tamalin [41], as described in the following section. These anatomical findings suggest that cytohesin-2 regulates membrane trafficking and actin cytoskeleton remodeling at various endomembrane systems and the plasma membrane in neurons.

## 3. Protein–Protein Interaction Networks of the Cytohesin Family

Cytohesin-1 was initially identified as an interacting protein of β2 integrin [11], and an increasing number of cytohesin-interacting proteins have been identified, including scaffold/adaptor proteins [37,41,42,43,44,45,46,47,48,49,50,51,52,53,54,55,56,57,58,59,60], GTPases and their regulatory proteins [24,25,61,62,63,64,65,66,67,68,69], transmembrane proteins [11,70,71,72,73,74,75,76,77,78], and cytoskeletal proteins [79,80], a neurodegenerative disease-related protein [33], and others [81,82] as summarized in Table 1. These interactions are considered to play critical roles in the proper functioning of cytohesin–Arf signaling through several mechanisms.

Firstly, the interaction determines the spatial and temporal specificity of Arf activation by recruiting cytohesins to particular subcellular locations at appropriate timings. For example, tamalin, also known as GRP1-associated scaffold protein (GRASP), is a postsynaptic scaffold protein with multiple protein-interacting domains including a PSD-95/discs large/ZO-1 (PDZ) domain, proline-rich region, leucine zipper region, and C-terminal PDZ-binding motif [41,42]. Tamalin interacts directly with the N-terminal coiled-coil domain of cytohesin-2/3 and the C-terminal PDZ-binding motif of mGluR1/5 through its leucine zipper region and PDZ domain, respectively [41]. Similar perisynaptic localization of cytohesin-2 and mGluR1/5 at excitatory synapses suggests that tamalin anchors cytohesin-2 in the proximity of mGluR1/5 at the perisynapse [23,37,38,39,40]. On the other hand, Munc13-1 is a component of the presynaptic active zone [83] and interacts with the N-terminal coiled-coil domain of cytohesin-1 [60]. The ability of Munc13-1 to form a protein complex simultaneously with cytohesin-1 and the presynaptic SNARE (soluble N-ethylmaleimide-sensitive factor attachment protein receptor) complex, including syntaxin 1A/B, SNAP25, and synaptobrevin 2, suggests that Munc13-1 recruits cytohesin-1 to the presynaptic active zone [60]. Although many of the cytohesin-interacting proteins identified so far are able to interact with more than one cytohesin family member, pallidin is unique in its strict binding specificity to cytohesin-2 [37]. Pallidin is a subunit of the biogenesis of lysosome-related organelle complex-1 (BLOC-1), which is implicated in membrane trafficking and protein sorting from early endosomes to various compartments, such as lysosome-related organelles [84], late endosomes/lysosomes [85], and primary cilium [86], and the biogenesis and release of synaptic vesicles in presynaptic terminals [87,88,89]. We have previously shown that cytohesin-2 is colocalized with pallidin preferentially in early endosomes and that knockdown of cytohesin-2 or pallidin similarly reduced the density of early endosomes in the somatodendritic compartment of cultured hippocampal neurons [37]. Thus, it is plausible that pallidin may recruit cytohesin-2 to early endosomes, thereby controlling Arf-dependent membrane trafficking and protein sorting from early endosomes.

Secondly, the interaction determines the selectivity of downstream Arf signaling by connecting cytohesins to downstream effectors and cargo proteins. Accumulating evidence implicates Arf6 in Rac1-dependent actin cytoskeleton reorganization in various cellular processes, including cell spreading and migration [90], phagocytosis [91,92], epithelial tubulogenesis [93], neurite outgrowth [94], and the formation of dendrites and dendritic spines [27,95]. During hepatocyte growth factor-stimulated epithelial cell migration, tamalin directly interacts with cytohesin-2 and Dock180, a Rac-GEF, thereby coupling cytohesin-2-dependent Arf1/6 activation to Dock180-dependent downstream Rac1 activation that leads to the actin cytoskeleton reorganization, suggesting that the interaction of cytohesin-2 with tamalin promotes crosstalk between Arf6 and Rac1 [96]. In addition, the connector enhancer of KSR (CNK) protein family, consisting of three members, CNK1, CNK2, and CNK3, functions as a scaffold protein with multiple protein-interacting domains including a sterile alpha motif, a conserved region in CNK, a PDZ domain, and a PH domain. All three CNK members interact with the coiled-coil domain of cytohesins [50,51,52,53]. Among these, CNK2, which is implicated in non-syndromic X-linked intellectual disability [97], was reported to form a multiprotein complex with cytohesin-2 and Rac/Cdc42 signaling molecules, such as Rac1 itself, Rac/Cdc42-GEFs (α-PIX and β-PIX), and Rac/Cdc42-GAP (Vilse/ARHGAP32), and to regulate dendritic spine morphogenesis in hippocampal neurons [51], suggesting that CNK2 may serve as a platform that connects Arf signaling to the Rac/Cdc42 pathway in the dendritic spine. Because Arf6 regulates dendritic branching and dendritic spine formation partly in a Rac1-dependent manner in cultured hippocampal neurons [27,95], future studies are needed to clarify the possible involvement of the multiprotein complex containing cytohesin-2 and Rac regulators via tamalin or CNK2 in neuronal functions.

Thirdly, the interaction plays a critical role in the regulation of the guanine nucleotide exchange activity of cytohesins. In resting conditions, cytohesins are strictly autoinhibited by intramolecular interaction of the linker region between the Sec7 and PH domains and the C-terminal helix/polybasic region with the Sec7 domain in a pseudosubstrate manner. Structural and biochemical analyses of cytohesin-2 and cytohesin-3 have revealed that the interaction of the PH domain with phosphoinositides, such as PI(4,5)P_2_ and PI(3,4,5)P_3_, and GTP-bound active Arf6 recruits cytohesins to the plasma membrane and induces the conformational rearrangement that relieves the autoinhibition, thereby leading to a fully active state of guanine nucleotide exchange activity toward Arfs [24,98,99,100]. Furthermore, Stalder et al. [101] demonstrated that the activity of cytohesins is modulated by a positive feedback loop, in which GTP-bound Arfs produced by cytohesins further activate cytohesins themselves through the interaction with the PH domain, thereby allowing the sustained activation of cytohesins and amplification of Arf signaling. Positive feedback loops of signaling pathways are considered to operate in various neuronal processes, as exemplified by the PI3K/Cdc42/Par complex/Rac-GEF/Rac1 pathway in axon specification during neuronal polarization [102], and the protein kinase C/extracellular signal-regulated kinase (ERK)/cytosolic phospholipase A_2_ pathway in cerebellar long-term depression (LTD), a form of synaptic plasticity related to motor learning [103,104]. The relationship of the unique activation properties of cytohesins with such neuronal functions remains to be clarified in future studies.

## 4. Roles of the Cytohesin Family in the Neuronal Development

During neuronal development, once neural progenitor cells complete the final cell division, neurons start to differentiate and become polarized by forming two distinct types of processes, axons and dendrites [105]. Axons then extend and reach their final targets by continuously making decisions as to whether they extend or retract through the interaction of the axon growth cone with various guidance cues, such as diffusible soluble chemical signals, cell surface molecules, and extracellular matrix. Neurite outgrowth and pathfinding are required for coordinated actions of the cytoskeleton and membrane trafficking, which not only provide the driving force for the growth cone to extend or retract but also supply membrane lipids and proteins, such as cell adhesion molecules and receptors, to the surface of expanding neurite tips [106,107]. There is accumulated evidence for the importance of the cytohesin–Arf pathway in the formation of axons and dendrites through actin cytoskeleton dynamics and membrane trafficking (Table 2).

### 4.1. Axonal Outgrowth

Jareb and Banker [110] demonstrated that treatment with BFA, a selective inhibitor of the GBF1 and BIG Arf-GEF subfamilies, reduced the number of polarized neurons with a single axon and inhibited axonal elongation, providing the first evidence for the importance of Arf-dependent membrane trafficking in neuronal polarization and neurite outgrowth. Hess et al. [111] subsequently demonstrated that Arfs are enriched in the axonal growth cone, a highly motile actin-rich structure at the tip of growing neurite that facilitates neurite outgrowth in response to environmental guidance cues, suggesting that Arfs control axonal development through membrane trafficking not only derived from the Golgi apparatus but also locally in the growth cone.

On the basis of this research background, Hernadez-Deviez et al. [28] provided the first evidence for the functional involvement of the cytohesin-2–Arf6 pathway in axonal elongation and branching. Overexpression of a catalytically inactive cytohesin-2 mutant (cytohesin-2^E156K^) resulted in marked increases in total axonal length and branching points of cultured hippocampal neurons, which were suppressed by coexpression of constitutively active Arf6 mutant (Arf6^Q67L^), suggesting that cytohesin-2 functions as an upstream activator of Arf6 in axonogenesis. They further identified PIP5Kα as a downstream effector of Arf6 in the axonal development, based on the findings that the axonal phenotypes induced by cytohesin-2^E156K^ or constitutively inactive Arf6 mutant (Arf6^T27N^) were suppressed by coexpression of wild-type PIP5Kα. The same group also demonstrated that overexpression of cytohesin-2^E156K^ or Arf6^T27N^ resulted in the depletion of Mena (mammalian-enabled), an actin regulatory protein of the Ena/VASP family, from the growth cone [28] and that overexpression of cytohesin-2^E156K^, Arf6^T27N^, or Arf6 mutant (Arf6^N48R^) that is defective in activating PLD induced the redistribution of a subset of endosomes labeled by endotubin, a glycoprotein associated with apical early endosomes, to the plasma membrane along the axon and at the growth cone [108]. Together, these series of studies suggest that the cytohesin-2–Arf6 pathway negatively regulates axonal outgrowth in hippocampal neurons by coordinating actin cytoskeleton remodeling and endosomal trafficking through PIP5K- and PLD-mediated lipid metabolism and signaling in the axon and growth cone. However, the results obtained by overexpression of cytohesin and Arf mutants should be interpreted carefully in terms of their specificity. Because four cytohesin paralogs are highly homologous and share various interacting proteins, it is likely that cytohesin-2^E156K^ may also suppress other cytohesins in a dominant negative manner. Indeed, we have demonstrated that knockdown of cytohesin-2 did not have significant effects on axonal length or tip number in hippocampal neurons [37]. The discrepancies in axonal phenotypes between overexpression of cytohesin-2^E156K^ and knockdown of cytohesin-2 may be explained by nonspecific inhibition of other cytohesins by overexpression of cytohesin-2^E156K^. Therefore, further studies are necessary to determine which cytohesin family members are responsible for axonal outgrowth by silencing individual cytohesins one by one.

The findings obtained from previous studies using the mouse neuroblastoma cell line N1E-115 will cast some light on the mechanisms underlying cytohesin–Arf6-dependent neurite outgrowth. N1E-115 cells have been extensively studied as a useful model for neuronal differentiation, because of their ability to differentiate into neuron-like phenotypes in response to various stimuli, such as serum deprivation, dibutyryl cyclic AMP, and valproic acid [112,113,114]. Yamauchi’s group identified actinin-1 and coiled-coil domain-containing protein 120 (CCDC120) as novel partner proteins for cytohesin-2 that are required for Arf6-dependent neurite outgrowth in N1E-115 cells [54,80]. Knockdown of actinin-1 or CCDC120 inhibits the neurite outgrowth and Arf6 activation induced by serum deprivation compared with the control. During neurite extension, actinin-1 is colocalized with cytohesin-2 in the growth cone [80], whereas CCDC120 recruits cytohesin-2 to transporting vesicles and is transported with cytohesin-2 along the neurite in an anterograde direction toward the growth cone [54]. These findings suggest that cytohesin-2 accumulates in the growth cone and controls Arf6 activation-dependent actin cytoskeleton dynamics in the growth cone through the interaction with actinin-1 and CCDC120 during neuronal differentiation in N1E-115 cells. It remains to be tested whether this model is applicable to the outgrowth of axons and/or dendrites in neurons.

### 4.2. Pathfinding

Commissural neurons send their axons to the contralateral side of the CNS to coordinate activity on both sides of the body. In the developing spinal cord, axons of dorsal commissural neurons initially grow toward the ventral midline under the guidance of a diffusible long-range chemoattractant, netrin, which is secreted by floor plate cells at the ventral midline [115,116]. Upon reaching the floor plate, they switch their responsiveness from netrin to chemorepellents secreted by the floor plate cells, such as Slit and semaphorin, thus exit the floor plate, and extend rostrally toward their final targets without re-crossing the midline [117].

Kinoshita-Kawada et al. [29] presented a novel mechanism by which the cytohesin–Arf6-dependent endocytic and recycling pathways control the switch of the responsiveness of commissural axons to Slit during midline crossing. They first found that endosomal trafficking mediated by Arf6, Rab5, and Rab11 small GTPases is required for repulsive response of commissural axons to Slit by increasing and maintaining the surface level of Robo1, a receptor for Slit, at distal axons of dorsal commissural neurons upon Slit stimulation. Consistent with this, Arf6 knockout mice exhibit a marked defect in midline crossing of commissural axons in the spinal cord, which is a similar phenotype observed in mutants lacking Robo1. They also found that cytohesins form a protein complex with Robo1 and activate Arf6 in commissural neurons upon stimulation with Slit. Furthermore, they demonstrated that knockdown of cytohesin-1 or cytohesin-3 in commissural neurons suppressed the response of commissural axons to Slit by reducing Slit-dependent Robo1 surface expression, thereby resulting in axon stalling within the floor plate without exiting the midline. On the other hand, knockdown of cytohesin-2 increased Robo1 levels and enhanced the repulsive response of commissural axons to Slit, resulting in axon stalling before entering the floor plate. These findings suggest that cytohesin-1/3 and cytohesin-2 play distinct roles in switching of the responsiveness of commissural axons to Slit during midline crossing: before axons reach the floor plate, cytohesin-2 suppresses the repulsive axonal response to Slit by downregulating surface Robo1, thereby allowing axons to enter the midline. Once axons reach the midline, cytohesin-1 and cytohesin-3 upregulate surface Robo1 levels in axon growth cones upon Slit stimulation through Arf6-dependent trafficking of Robo1 to the plasma membrane, thereby allowing axons to become repulsive to Slit and to exit the midline.

Midline crossing of axonal projections is present at various brain regions, such as the corpus callosum, anterior and posterior commissures, optic chiasm, and pyramidal decussation. Notably, neuron-specific Arf6 conditional knockout (cKO) mice exhibit a marked reduction in the size of the corpus callosum, the largest commissural fiber tract between the cerebral hemispheres [118]. Although this phenotype was reported to be attributable to reduced myelination caused by impaired migration of oligodendrocyte precursor cells through Arf6-dependent secretion of guidance factors such as fibroblast growth factor-2 from neurons [118], the possibility that the cytohesin–Arf pathway mediates axonal pathfinding and midline crossing in the corpus callosum and other fiber tracts through endosomal trafficking of axon guidance receptors is worth testing in future studies.

### 4.3. Dendritic Development

Hernadez-Deviez et al. [27] first demonstrated that overexpression of cytohesin-2^E156K^ or Arf6^T27N^ increased the tip number of dendrites in cultured rat hippocampal neurons. This dendritic phenotype was blocked completely by coexpression of Arf6^Q67L^ and partially by that of wild-type Rac1. Therefore, cytohesin-2 negatively regulates Arf6-dependent dendritic arborization in part through the Rac1-dependent downstream pathway, which differs from cytohesin-2–Arf6-mediated axonogenesis in downstream signaling pathways [28]. However, these results also include the same issue about the specificity of cytohesin-2 and Arf6 mutants as described above. In fact, we have demonstrated that knockdown of cytohesin-2 reduces total dendritic length without affecting dendritic tip number in hippocampal neurons [37]. Because hippocampal neurons express at least three *cytohesins*, except *cytohesin-4*, at the mRNA level [36], further studies are needed to determine which cytohesin family members are responsible for Arf6-dependent dendritic development in hippocampal neurons by silencing individual cytohesins one by one.

The ubiquitin proteasome system is implicated in diverse neuronal processes, including neuronal migration, the development of axons and dendrites, synapse formation, and synaptic plasticity, through the regulation of protein degradation [119,120]. Han et al. [109] demonstrated that dorsal telencephalon-specific cKO mice for RING box protein 2 (RBX2), an essential component of the Cullin-RING E3 ubiquitin ligase 5 (CRL5) complex, exhibited a defect in apical dendritic development as well as neuronal positioning of hippocampal pyramidal cells. A further quantitative proteomic analysis identified that the expression of Arf-like protein (Arl) 4C, cytohesin-1/3, Arf6, and FERM domain-containing protein 4A (FRMD4A) were post-translationally upregulated with elevated GTP-bound active Arf6 in the hippocampus of RBX2 cKO mice [109]. Simultaneously silencing of Cullin 5 of the CRL5 complex with cytohesin-1 or Arf6 using in utero electroporation resulted in the failure of transfected hippocampal pyramidal cells to extend their apical dendrites into the stratum lacunosum-moleculare, although silencing of Cullin 5 alone did not have such an effect on dendritic extension, suggesting that the defect in the extension of apical dendrites is dependent on the loss of cytohesin-1 or Arf6. Furthermore, silencing of Arf6 and Arl4C individually blocked the ability of hippocampal pyramidal cells to extend their apical dendrites into the stratum lacunosum-moleculare [109]. Because Arl4 was reported to function upstream of cytohesin-1 by recruiting it to the plasma membrane to activate its guanine nucleotide exchange activity [61,62], it is hypothesized that the Arl4C-cytohesin-1-Arf6 signaling cascade controls the apical dendritic development of hippocampal pyramidal cells under the control of CRL5-mediated ubiquitin proteasome system. Furthermore, cytohesin-1 was reported to form a protein complex with FRMD4A to regulate Arf6-dependent formation of primordial adherence junctions during epithelial polarization [49]. Therefore, the functional significance of the interaction among cytohesin-1, Arf6, Arl4C, and FRMD4A in the development of hippocampal apical dendrites remains to be elucidated in future studies.

## 5. Roles of the Cytohesin Family in Mature Neuronal Functions

Although evidence for the importance of the cytohesin–Arf pathway in developing neuronal functions has been accumulated as described above, its roles in mature neuronal functions are largely unknown. Here, we summarize previous findings on the cytohesin–Arf pathway in mature neuronal functions obtained mainly by pharmacological blockade of cytohesins with SecinH3, suggesting their implication in both pre- and post-synaptic functions (Table 2).

### 5.1. Presynaptic Functions

The fidelity of neurotransmission depends on continuous cycling of synaptic vesicles in the presynaptic terminals, which consists of various steps including the biogenesis, docking, priming, fusion, retrieval, and reformation of synaptic vesicles. Recent evidence suggests that synaptic vesicle cycling is structurally and functionally related to the endolysosomal trafficking system [121]. The first clue implicating Arfs in regulated secretion came from a study demonstrating that functional interference of Arf6 with a synthetic myristoylated peptide corresponding to the N-terminal region of Arf6 inhibited calcium-stimulated exocytosis of large dense-core vesicles in permeabilized chromaffin cells [122]. Subsequent studies using chromaffin cells and PC12 cells, a neuroendocrine cell line derived from rat pheochromocytoma, revealed that Arf6 is implicated in calcium-regulated exocytosis of dense-core vesicles through downstream activation of PLD and PIP5Kγ [123,124]. Concerning an upstream activation mechanism for Arf6, microinjection of anti-cytohesin-2 antibodies into the cytosol of permeabilized chromaffin cells inhibited calcium-evoked catecholamine secretion and activation of PLD. Furthermore, coexpression of wild-type cytohesin-2 with human growth hormone enhanced depolarization-induced secretion of growth hormone, whereas coexpression of cytohesin-2^E156K^ suppressed it in PC12 cells, indicating that cytohesin-2 functions as an upstream GEF for Arf6 in calcium-regulated secretion of large dense-core vesicles in neuroendocrine cells [125]. Consistent with the mechanistic similarity of exocytosis between large dense-core vesicles and synaptic vesicles, overexpression of wild-type cytohesin-1 in cultured *Xenopus* spinal neurons and *Aplysia* pedal ganglion sensory neurons was reported to increase basal synaptic transmission at neuromuscular junctions [30] and sensory-to-motor synapses [31], respectively. Furthermore, cytohesin-1 was shown to interact with Munc13-1 [60], a component of the presynaptic active zone [83] that is essential for the fusion competence of glutamatergic synaptic vesicles [126]. These findings suggest that the cytohesin-2–Arf6 pathway may control the exocytosis of synaptic vesicles at the presynaptic active zone. In addition to exocytosis of synaptic vesicles, Arf6 was also implicated in clathrin-mediated endocytosis of the synaptic vesicle membrane through the downstream activation of PIP5Kγ and subsequent recruitment of coat components including clathrin, AP-2, and AP180 [127], although the functional involvement of cytohesin in the retrieval of synaptic vesicles from the plasma membrane remains to be elucidated.

Tagliatti et al. [32] elegantly demonstrated that knockdown of Arf6 induces characteristic presynaptic phenotypes, including a decrease in synaptic vesicles, increase in morphologically docked vesicles at the active zone, and accumulation of endosome-like cisternae in the presynaptic terminal of cultured hippocampal neurons, suggesting that Arf6 regulates the readily releasable pool of synaptic vesicles and the recycling pathway of retrieved synaptic membrane to reform synaptic vesicles. Because treatment of cultured hippocampal neurons with SecinH3 had the same effects on presynaptic structures as Arf6 knockdown, it is plausible that cytohesin functions as an upstream GEF in Arf6-mediated presynaptic functions of hippocampal neurons. Further studies are necessary to determine which cytohesin family members are responsible for Arf6-mediated synaptic vesicle recycling, and how cytohesins are activated to initiate Arf6 signaling in presynapses.

### 5.2. Postsynaptic Functions

Arf1 and Arf6 are implicated in hippocampal synaptic plasticity, particularly LTD, through actin cytoskeleton remodeling and endosomal trafficking of α-amino-3-hydroxy-5-methyl-4-isoxazolepropionate-type glutamate receptors (AMPARs) at postsynapses [128,129]. Although two BRAG family members, BRAG1 and BRAG2, were shown to mediate the endocytosis of AMPARs as upstream Arf-GEFs during hippocampal LTD [128,130,131], the roles of cytohesins in postsynaptic functions have not been clarified yet. A clue to understand the postsynaptic function of cytohesins is previous findings on the association of cytohesin-2 with mGluR1/5 [23,41]. As described in Section 3, Kitano et al. [41] demonstrated that cytohesin-2 is enriched in the postsynaptic membrane fraction and forms a protein complex with mGluR1/5 through the interaction with tamalin. In addition to mGluR1/5 and cytohesins, tamalin interacts with various synaptic scaffold proteins including PSD-95, SAPAP1/3, S-SCAM, Mint2, and CASK [132], suggesting its participation in receptor clustering, trafficking, and intracellular signaling. Indeed, knockdown of tamalin was recently shown to inhibit mGluR1/5-stimulated internalization of mGluR1/5 and AMPAR subunit GluA1 in cultured hippocampal neurons [133]. Furthermore, pre-administration of a cell-permeable peptide corresponding to the PDZ-binding motif of mGluR1, which was designed to competitively block the interaction between mGluR1/5 and tamalin into the lateral ventricle, was shown to inhibit the expression of LTD induced by low-frequency stimulation as well as the facilitation of short-term depression into LTD elicited by pharmacological activation of the group I mGluR agonist, 3,5-dihydroxyphenylglycine (DHPG), in the CA1 dendritic area of hippocampal organotypic slice cultures, without any effects on long-term potentiation or basal synaptic transmission [134]. Notably, exogenous expression of a C-terminally truncated tamalin mutant lacking the ability to interact with cytohesins inhibited the cell-surface expression of transfected mGluR1a in COS-7 cells and the distribution of endogenous mGluR5 along neurites in cultured hippocampal neurons [41]. These findings led us to speculate that cytohesins, particularly cytohesin-2, may control mGluR1/5-dependent synaptic transmission and plasticity at the postsynaptic site through Arf-dependent trafficking of mGluR1/5 and AMPARs, which needs to be clarified in future studies.

## 6. Pathological Roles of the Cytohesin Family in the CNS

In this section, we review recent findings on the functional involvement of the cytohesin family in pathological conditions that are related to chronic pain and neurological diseases in the CNS (Table 3).

### 6.1. Chronic Pain

Chronic pain is one of the most prevalent complaints of patients in clinical settings and associated with clinical manifestations, such as spontaneous pain, augmented pain responses to noxious stimuli (hyperalgesia), and pain generated by innocuous stimuli (allodynia) [135,136]. Chronic pain develops and persists in part by increased excitability of dorsal horn neurons in the spinal cord, which represents a form of neural plasticity, referred to as central sensitization, in response to nociceptive stimuli caused by various insults, such as inflammation and neuropathic injury [137]. In the spinal cord, glutamate mediates and modulates nociceptive transmission at excitatory synapses through two classes of glutamate receptors: ionotropic glutamate receptors and mGluRs [138,139]. Like other forms of synaptic plasticity, glutamate receptor-mediated signaling pathways are implicated in the development and maintenance of central sensitization [140,141]. However, our understanding of the detailed molecular mechanisms underlying central sensitization is still far from complete. We have recently provided the first evidence for the functional involvement of the cytohesin-2–Arf6 pathway in central sensitization [23].

In the superficial dorsal horn of the spinal cord, cytohesin-2 is expressed abundantly in subsets of excitatory interneurons and projection neurons, but not in inhibitory neurons or glia. Immunoelectron microscopic analyses of dorsal horn neurons revealed that cytohesin-2 is associated with the plasma membrane, and membrane vesicles in various subcellular compartments, particularly in the somatodendritic compartment. In excitatory synapses of dorsal horn neurons, cytohesin-2 is localized at the postsynaptic membrane, with accumulation preferentially around the edge of the postsynaptic density [23], which is reminiscent of perisynaptic localization of mGluR5 [39,40]. Indeed, cytohesin-2 forms an immunoprecipitable protein complex with mGluR5 in the synaptosomal fraction of the spinal cord [23], suggesting that cytohesin-2 is strategically located at the perisynapse to regulate mGluR5-mediated signaling in the postsynapse of dorsal horn neurons.

Central nervous system-specific cytohesin-2 cKO mice exhibited reduced mechanical allodynia following either intradermal injection of complete Freund’s adjuvant into the footpad (inflammatory pain model) or partial ligation of the sciatic nerve (neuropathic pain model) without differences in basal mechanical pain sensitivity compared with the control [23]. In inflammatory and neuropathic pain models, both Arf1 and Arf6 were activated with a peak between 6 and 12 hours following stimulation. Furthermore, acute inhibition of the catalytic activity of cytohesins in the spinal cord by intrathecal administration of SecinH3 reduced mechanical allodynia and activation of Arf6, but not Arf1, in both pain models [23]. These findings not only exclude the possibility of the developmental defects in nociceptive transmission caused by the loss of cytohesin-2 in cKO mice but also suggest that cytohesin-2 mediates the development of central sensitization in the spinal cord through the activation of Arf6. Furthermore, the anti-allodynic effect of SecinH3 in both inflammatory and neuropathic pain models suggests the cytohesin-2–Arf6 pathway to be a potential broad-spectrum therapeutic target for chronic pain.

What is the mechanism by which the cytohesin-2–Arf6 pathway mediates central sensitization? The physical association of cytohesin-2 with mGluR5 in the spinal cord motivated us to further examine the functional involvement of cytohesin-2 in mGluR1/5-dependent nociceptive behavior. Pharmacological stimulation of spinal mGluR1/5 by intrathecal injection of DHPG induced mechanical allodynia, which persisted for at least 7 days in control mice. In contrast, cytohesin-2 cKO mice exhibited markedly reduced mechanical allodynia induced by DHPG, suggesting that cytohesin-2 mediates the development of chronic pain downstream of mGluR1/5 in the spinal cord [23]. ERK1 and ERK2 (ERK1/2) are one of the major downstream effectors of Arf6 signaling in various physiological and pathological cellular processes such as epithelial tubule development [93], and cancer cell migration and invasion [142,143]. Because ERK1/2 also contribute to the development of central sensitization in spinal cord dorsal horn neurons [144,145,146], we examined whether ERK1/2 were activated by stimulation of spinal mGluR1/5 with DHPG in the dorsal horn of cytohesin-2 cKO mice and found that the phosphorylation of ERK1/2 induced by intrathecal DHPG administration was markedly attenuated in the spinal cord of cytohesin-2 cKO mice [23]. Together, it is plausible that the cytohesin-2–Arf6 pathway contributes to mGluR1/5-dependent central sensitization through downstream ERK1/2 activation. Notably, the post-embedding immunoelectron microscopy of excitatory synapses of dorsal horn neurons revealed that the peak of synaptic distribution of mGluR5, but not the cognate G protein Gαq/11/14, was shifted from the perisynapse to the postsynaptic density in cytohesin-2 cKO mice [23]. Although the mechanism underlying the change in mGluR5 perisynaptic positioning in cytohesin-2 cKO mice remains to be elucidated, it is possible that cytohesin-2 may regulate the perisynaptic positioning of mGluR5 in dorsal horn neurons through Arf6-dependent membrane trafficking and/or actin cytoskeletal remodeling, thereby efficiently coupling mGluR5 to downstream ERK1/2 signaling that leads to central sensitization.

### 6.2. Neurodegenerative Disease

#### 6.2.1. ALS

ALS is a devastating neurodegenerative disease defined by progressive degeneration and loss of both upper and lower motor neurons. An increasing number of mutations in genes encoding various proteins, such as superoxide dismutase 1 (SOD1), transactive response DNA-binding protein 43 (TDP-43), and chromosome 9 open reading frame 72 (C9ORF72), have been identified to cause ALS [147]. Although the pathogenic mechanisms are not completely understood, recent evidence indicates that the mutations confer a loss of function and/or toxic gain of function, which result in the impairment of various membrane trafficking processes such as autophagy–lysosomal pathway and axonal transport, eventually leading to neuronal cell death [148].

Two recent studies using SecinH3 have provided evidence for the cytohesin–Arf pathway as a potential therapeutic target for ALS (Table 3). Mutations in the gene encoding SOD1, a ubiquitous antioxidant enzyme, were first identified as a cause of ALS [149]. Such mutations are the second most common in European ALS populations and the most frequent in Asian ALS populations [150]. Zhai et al. [33] demonstrated that pharmacological inhibition of cytohesin activity with SecinH3 or knockdown of cytohesins has a protective effect on neurotoxicity induced by expression of mutant human SOD1 (SOD1^G85R^), which is prone to misfolding and aggregation, in cultured spinal motor neurons by alleviating endoplasmic reticulum (ER) stress and stimulating autophagic flux, thereby leading to the clearance of mutant SOD1 proteins. The stimulatory effect of SecinH3 on the degradation of mutant and wild-type SOD1 proteins was blocked by overexpression of constitutively active mutant of Arf1 (Arf1^Q71L^) or Arf5 (Arf5^Q71L^), but not Arf6^Q67L^, suggesting that the activation of Arf1 and Arf5 by cytohesins is responsible for SOD1 mutant-induced neurotoxicity. Of note, cytohesins interact with both wild-type SOD1 and SOD1^G85R^ with a higher affinity to SOD1^G85R^, although the interaction does not affect their guanine nucleotide exchange activity [33]. Thus, the interaction of cytohesins with mutant SOD1 may mislocalize cytohesins in neurons expressing mutant SOD1, thereby leading to aberrant Arf activation in inappropriate subcellular compartments and the dysfunction of normal Arf-mediated processes. Hu et al. [34] subsequently demonstrated that SecinH3 also has protective effects against neurotoxicity induced by expression of mutant TDP-43 (TDP-43^Q331K^), which is associated with ALS and frontotemporal dementia (FTD), by reducing ER stress-mediated apoptosis and promoting the autophagic flux in SH-SY5Y cells, a human neuroblastoma cell line. Together, the neuroprotective effects of SecinH3 against neurotoxicity induced by different mutant proteins associated with ALS pathogenesis suggest that the cytohesin–Arf pathway is a promising therapeutic target for ALS.

The expansion of the hexanucleotide GGGGCC repeat in the first intron of the C9ORF72 gene is the most common causative mutation in familial ALS and FTD in Caucasian populations [151,152]. The mutation appears to cause ALS and FTD through loss of function due to reduced C9ORF72 protein expression and/or gain of toxic function due to the aggregation of RNA and dipeptide repeat proteins [153]. It is worth noting that recent findings suggest the functional relationship between C9ORF72 and the Arf pathway. Sivadason et al. [154] demonstrated that knockdown of C9ORF72 reduced the axonal length and growth cone size in primary mouse embryonic motor neurons, whereas overexpression of C9ORF72 had the opposite effects, suggesting that C9ORF72 positively regulates axonal outgrowth. Subsequent mass spectrometry-based proteomics of immunoprecipitates from neuroblastoma NSC-34 cells overexpressing human C9ORF72 identified cofilin, Arp2/3, coronin, Arf1, and Arf6 as novel C9ORF72 interactors [154]. Among these, cofilin is an actin-binding protein that positively regulates axonal growth cone dynamics and axonal extension by severing and depolymerizing aged actin filaments, and its activity on actin dynamics is negatively regulated by LIM domain kinase (LIMK) 1/2-mediated phosphorylation downstream of Rac1 [155]. Knockdown of C9ORF72 enhanced the phosphorylation of cofilin with reduced axonal actin dynamics in cultured mouse motor neurons. Consistent with this, enhanced phosphorylation of cofilin with reduced endogenous C9ORF72 level was also observed in postmortem cerebellar samples from ALS patients with C9ORF72 repeat expansion. Notably, knockdown of C9ORF72 also increased the levels of active GTP-bound Arf6 and Rac1, suggesting that C9ORF72 negatively regulates the activation of the Arf6-Rac1 signaling pathway that leads to LIMK1/2-mediated phosphorylation of cofilin. In fact, knockdown of Arf6 blocked the enhanced Rac1 activation observed in induced pluripotent stem cell-derived motor neurons from ALS patients with C9ORF72 repeat expansion, and coexpression of Arf6^T27N^ in C9ORF72-knockdown mouse motor neurons rescued the inhibitory effect of C9ORF72 knockdown on axonal elongation. Together, they proposed a model in which C9ORF72 regulates cofilin-mediated axonal extension and growth cone dynamics by inhibiting the Arf6-Rac1-LIMK1/2 signaling pathway. Consistent with the finding that the depletion of C9ORF72 leads to Arf6 activation, Su et al. [156] demonstrated that C9ORF72 functions as a GAP for Arf1 and Arf6 by forming a protein complex with SMCR8 (Smith-Magenis chromosome region 8) and WDR41 (WD repeat-containing protein 41). Therefore, it is fascinating to speculate that aberrant activation of Arf6 caused by the loss of C9ORF72 may be related to the pathophysiology of ALS. It remains to be clarified in future studies whether the cytohesin–Arf pathway is involved in pathophysiological processes of ALS caused by an abnormal C9ORF72 repeat expansion.

#### 6.2.2. AD

AD is the most common form of dementia with the increasing incidence and prevalence with age. The pathological hallmarks of AD are cortical and hippocampal atrophy caused by the loss of neurons and synapses with the presence of two types of abnormal structures: neurofibrillary tangles consisting of an aggregation of misfolded tau proteins, and amyloid plaques consisting of a fibrillary aggregation of amyloid-β (Aβ) peptides derived from amyloid precursor protein (APP).

Tau is a microtubule-associated protein that stabilizes microtubules to ensure proper cytoskeleton organization and membrane transport in the axon [157]. In pathological conditions, it is hyperphosphorylated, misfolded, and aggregates into insoluble bundles of paired helical filaments, which contribute to the progression of diseases, referred to as tauopathy, including AD, FTD, Pick’s disease, progressive supranuclear palsy, and corticobasal degeneration [158]. Consistent with the autopsy findings that the accumulation of tau inclusion in AD propagates in a stereotypical pattern along neuroanatomically interconnected brain regions [159], multiple lines of evidence indicate that pathogenic misfolded tau proteins are secreted into the extracellular space with the ability to transfer from affected neurons to neighboring healthy neurons and to promote misfolding and aggregation of normal tau molecules in healthy neurons that have internalized pathogenic tau [160,161]. Although pathogenic tau proteins are transferred via various mechanisms, including direct secretion through the plasma membrane, secretion in exosomes or ectosomes, and direct cell–cell transfer via tunneling nanotubes [162], our understanding of the detailed molecular mechanisms for tau secretion, which facilitates the development of therapeutic strategies to prevent the disease propagation, is still far from complete.

Yan et al. [35] established a novel sensitive live-cell assay for monitoring tau dimer secretion using fusion proteins of tau with two complementary fragments of the humanized Guassia princeps luciferase reporter protein. They found that knockdown of FRMD4A, whose gene polymorphisms are a risk factor for AD [163], reduced the secretion of tau dimer from transfected HEK293T cells into the culture medium, whereas overexpression of FRMD4A enhanced the tau secretion in a dose-dependent manner. Because FRMD4A regulates Arf6-dependent formation of primordial adherence junctions during epithelial polarization by forming a protein complex with cytohesin-1 and the Par3/Par6/atypical protein kinase C complex [49], the authors then examined the functional involvement of the cytohesin–Arf6 pathway in FRMD4A-induced tau secretion [35]. As a result, SecinH3 treatment suppressed FRMD4A-induced tau secretion, and overexpression of wild-type Arf6 or Arf6^Q67L^ significantly enhanced tau secretion in HEK293T cells. However, in contrast to what was observed in a non-neuronal cell line, knockdown of FRMD4A or SecinH3 treatment promoted the secretion of endogenous tau in cultured mouse cortical neurons [35]. Although the reason for the difference in the effects of FRMD4A knockdown and SecinH3 treatment on tau secretion between neurons and HEK293T cells is unknown at present, these findings suggest that the expression level of FRMD4A positively or negatively correlates with the activity of tau secretion depending on cell types. Further studies are necessary to examine the possibility that cytohesins control the propagation of tau pathology in AD through Arf6-dependent tau secretion. It should be noted that other misfolded and aggregated pathogenic proteins associated with neurodegenerative diseases, such as Aβ for AD, TDP-43 and SOD1 for ALS, and α-synuclein for Parkinson’s disease, are also proposed to spread in a prion-like manner along neural connections [160,161]. Thus, it will be interesting to examine whether the cytohesin–Arf pathway plays a general role in the secretion and propagation of pathogenic protein aggregation in neurodegenerative diseases.

Another pathological hallmark of AD is the accumulation of amyloid plaques composed primarily of the Aβ peptide, which is generated by sequential proteolytic cleavage of APP by β- and γ-secretases. The β-site APP-cleaving enzyme 1 (BACE1) is the major β-secretase that catalyzes the limiting step in the production of the Aβ [164]. Because both APP and BACE1 are transmembrane proteins that are transported between the plasma membrane and intracellular membrane-bounded organelles, such as the trans-Golgi network and endosomes, intracellular trafficking of both molecules is believed to be a critical regulatory mechanism that controls the timing and duration for processing and secretion of Aβ [165]. Two independent lines of evidence suggest the implications of the Arf-dependent membrane trafficking in AD through APP processing. Firstly, Sannerud et al. [166] demonstrated that Arf6 regulates the sorting of BACE1 to early endosomes after internalization, thereby regulating APP processing in endosomes. Secondly, Tang et al. [167] demonstrated that Arf6 mediates the internalization of APP from the plasma membrane to the lysosome via macropinocytosis in NS56 neuroblastoma cells, thereby regulating APP processing and Aβ production in the lysosome. They further demonstrated by immunostaining that Arf6 protein levels were increased in the postmortem hippocampus of patients with AD in parallel with disease progression [167]. These findings suggest that dysfunction of the Arf6 pathway is associated with the pathogenesis of AD by controlling the Aβ production through intracellular trafficking of APP and BACE1. Thus, it is challenging to examine whether the cytohesin–Arf pathway is involved in pathological processes of AD and serves as a potential therapeutic target in future studies.

## 7. Concluding Remarks

Recent advances in molecular biological techniques and the development of SecinH3 have enhanced our understanding of multifaceted roles of the cytohesin–Arf pathway in physiological and pathological functions in neurons. However, there still remain many key questions to be answered to understand the neuronal functions of the cytohesin–Arf pathway.

Firstly, although pharmacological approaches with SecinH3 have demonstrated the functional involvement of cytohesins as a whole in various neuronal functions, we are still ignorant of the functional redundancy and nonredundancy among the cytohesin family members. The four cytohesin family members are highly homologous with overlapping interacting proteins (Table 1) and mRNA expression patterns [36]. Therefore, experimental approaches using depletion of individual cytohesins at cell and animal levels need to complement the results obtained using SecinH3 and to clarify neuronal functions specific for individual cytohesins.

Secondly, our knowledge of the in vivo substrates of cytohesins is incomplete, as described in Section 1. Although cytohesins show guanine nucleotide exchange activity toward all Arf members, most of the previous in vivo studies analyzed only Arf1 and Arf6 as a substrate. Thus, it is important to identify the physiological substrate of cytohesins and the mechanism of how cytohesins discriminate the substrate depending on the cellular context. 

Thirdly, although previous in vitro biochemical analyses have revealed that cytohesins are subject to elaborate regulatory mechanisms for activation, including intramolecular autoinhibition, interactions with lipids and proteins, and post-translational modifications such as phosphorylation and dephosphorylation [24,61,62,98,99,100,101,168,169,170,171], the activation mechanisms in vivo are largely unknown in physiological and pathological processes in neurons. Recent elegant in vivo studies using transgenic mice revealed that the phosphorylation of cytohesins-1 and -2 by the Src family tyrosine kinase Fyn at positions 382 and 381, respectively, in their C-terminal autoinhibitory region is required for Arf6-dependent myelination in Schwann cells [169,171]. Further in vivo studies are necessary to unravel the activation mechanisms for cytohesins in neurons in physiological and pathological conditions. 

Fourthly, despite encouraging evidence for the cytohesin–Arf pathway as a potential therapeutic target for chronic pain [23] and neurological disorders [33,34], the general pharmacological inhibition of cytohesins with SecinH3 is unlikely to be clinically applicable, because of the fundamental biological importance of cytohesin–Arf-mediated membrane trafficking and cytoskeletal processes. In fact, treatment of mice with SecinH3 induced hepatic insulin resistance by altering the insulin-regulated expression of various genes related to gluconeogenesis, glycolysis, and fatty acid synthesis in the liver [26]. To develop strategies to pinpoint the signaling components that function in cytohesin–Arf signaling in specific neuronal processes, it is crucial to identify comprehensive protein–protein networks and regulatory mechanisms of the cytohesin–Arf pathway in individual neuronal processes.

Lastly, recent evidence suggesting the relationship between the cytohesin–Arf pathway and neurodegenerative disorders was obtained only by overexpression experiments in cell lines and primary neurons [33,34,35]. As a next step, it will be necessary to examine the functional involvement of the cytohesin–Arf pathway in the pathogenesis of neurodegenerative disorders using animal models of human neurological diseases.

We hope that this review will enhance this research field to obtain further exciting discoveries on the physiological and pathological importance of the cytohesin–Arf pathway in the nervous system in the future.

## Figures and Tables

**Table 1 ijms-23-05087-t001:** Cytohesin-interacting proteins.

Interacting Proteins	Proposed Functions	Specificity for Interaction *1	Interaction Domains in Cytohesin	Experimental Approaches *2	Refs
* **Scaffold/adaptor proteins** *					
Tamalin/GRASP	・ Trafficking and surface expression of group I mGluRs ・ Arf-to-Rac crosstalk by forming a protein complex with cytohesin-2 and Dock180 during epithelial cell migration ・ Neurotrophin-3-induced actin reorganization by forming a protein complex with cytohesin-2 and TrkCT1	Cyth-2, -3 (Cyth-1, -4: ND)	CC	Y2H, PD, IP(exo), IP(endo)	[41,42,43]
CASP/Cybr/CYTIP	・ β2 integrin-dependent cell adhesion in lymphocytes by sequestering cytohesins ・ Endosomal trafficking and sorting by forming a ternary complex with cytohesin and sorting nexin 27 in lymphocytes	Cyth-1, -2, -3 (Cyth-4: ND)	CC	Y2H, PD, IP(exo), IP(endo)	[44,45,46,47]
FRMD4A FRMD4B/GRSP1	・ Arf6-dependent formation of adherence junctions by recruiting cytohesin-1 to the Par complex in primordial adherence junctions during epithelial polarization	Cyth-1, -2, -3 (Cyth-4: ND)	CC	Y2H, PD, IP(exo), IP(endo)	[48,49]
CNK1	・ Insulin-dependent recruitment of cytohesins to the plasma membrane (PM) and facilitation of IRS1/phosphatidylinositol 3-kinase/Akt signaling through activation of the Arf-PIP5K pathway	Cyth-1, -2, -3 (Cyth-4: ND)	CC	MS(exo), IP(exo), IP(endo)	[50]
CNK2A/MAGUIN-1	・ Neurite outgrowth in NG108 cells and spine morphogenesis in hippocampal neurons by forming a multiprotein signaling complex including cytohesin, GIT1/2, Vilse/ARHGAP39, α/β-PIX, and PAK3/4	Cyth-2 (Cyth-1, -3, -4: ND)	CC	MS(exo), IP(exo), IP(endo)	[50,51]
CNK3/IPCEF1	・ Hepatocyte growth factor-dependent Arf6 activation and scattering/migration of MDCK cells	Cyth-1, -2, -3, -4	CC	Y2H, PD, IP(exo)	[52,53]
CCDC120	・ Recruitment of cytohesin-2 to transporting vesicles along neurites, and neurite outgrowth through Arf6 activation in N1E-115 cells	Cyth-2 (Cyth-1, -3, -4: ND)	CC	Y2H, PD, IP(exo), IP(endo)	[54]
Paxillin	・ Migration of preadipocyte 3T3-L1 cells through the activation of Arf6	Cyth-2 (Cyth-1, -3, -4: ND)	PB	PD, IP(exo), IP(endo)	[55,56]
β-Arrestin-1/2	・ Arf6-dependent endocytosis of β2-adernergic receptor upon ligand stimulation through the recruitment of cytohesin-2 to the PM ・ Calcium-sensing protein (CaSR)-stimulated cytoskeletal reorganization and PM ruffling through β-arrestin-1– cytohesin-2–Arf6–ELMO protein network ・ Angiotensin II type 1 receptor-stimulated cell migration through Arf6-dependent endocytosis and mitogen-activated protein kinase activation	Cyth-1, -2 (Cyth-3, -4: ND)	ND	IP(exo), PD	[57,58,59]
Munc13-1	・ Neurotransmitter release in the presynaptic axon terminal	Cyth-1 (Cyth-2, -3, -4: ND)	CC	Y2H, PD	[60]
Pallidin	・ Early endosome dynamics and dendritic growth of cultured hippocampal neurons	Cyth-2	CC	Y2H, PD, IP(exo), IP(endo)	[37]
* **GTPases and their regulators** *					
Arf6	・ Recruitment of cytohesins to the PM to activate Arf6	Cyth-2, -3 (Cyth-1 -4: ND)	PH	IP(exo)	[24]
Arl4	・ Recruitment of cytohesins to the PM to promote Arf6-dependent actin remodeling and cell migration	Cyth-1, -2, -3, -4	PH+PB	Y2H, PD, IP(exo)	[61,62]
ARD1	・ Recruitment GDP-ARD1 to exchange GDP for GTP as a substrate	Cyth-1 (Cyth-3, -4: ND)	Sec7	Y2H, PD	[25]
ARP	・ Negative regulation of Arf-dependent phospholipase D activation upon stimulation of muscarinic acetylcholine receptor-3 by preventing the recruitment of cytohesin to the PM	Cyth-1, -2 (Cyth-3, -4: ND)	Sec7	Y2H, PD	[63]
Gαq	・ Agonist-induced internalization of the thromboxane A2 receptor through the recruitment of cytohesin to the PM and Arf6 activation	Cyth-1, -2, -3 (Cyth-4: ND)	CC	PD, IP(exo)	[64,65]
Cytohesin	・ Homodimerization	Cyth-2 (Cyth-1, -3, -4: ND)	CC	IP(exo)	[66]
TBC1D10A/EPI64	・ Glucose-dependent endocytosis through Arf6 activation and recruitment of TBC1D10A to the PM to activate Rab27a	Cyth-2 (Cyth-1, -3, -4: ND)	PH	PD, IP(exo), IP(endo)	[67]
RLIP76	・ Cell spreading and migration by connecting activated R-Ras with the downstream cytohesin-2-Arf6 signaling	Cyth-2 (Cyth-1, -3, -4: ND)	ND	IP(exo)	[68,69]
** *Transmembrane proteins* **					
β2 integrin	・ LFA1-mediated adhesion to ICAM-1 in lymphocytes through inside-out signaling of β2 integrin ・ Negative regulation of Mac1-dependent adhesion, phagocytosis, and chemotaxis in neutrophils	Cyth-1, -3 (Cyth-2, -4: ND)	Sec7	Y2H, PD, IP(endo)	[11,70,71,72]
Insulin receptor	・ Arf1-dependent activation of phospholipase D upon insulin stimulation	Cyth-2 (Cyth-1, -3, -4: ND)	CC + PH	IP(exo)	[73]
V-ATPase a subunit	・ Membrane trafficking between early endosomes to late endosomes in renal proximal tubule epithelial cells through intraendosomal acidification-dependent recruitment of cytohesin-2 and Arf6 to early endosomes	Cyth-2 (Cyth-1, -3, -4: ND)	Sec7, (PH, PB)	M2H, PD, SPR, IP(endo)	[74,75]
A2A adenosine receptor	・ Agonist-induced sustained activation of mitogen-activated protein kinase through Arf6 activation	Cyth-2 (Cyth-1, -3, -4: ND)	PH	Y2H, PD, IP(exo) DRAP-FRET	[76]
EGFR	・ Modulation of EGFR activation	Cyth-2 (Cyth-1, -3, -4: ND)	Sec7	MST	[77]
Kaposin A	・ Human herpesvirus *kaposin A*-induced transformation of fibroblasts through the recruitment of cytohesin-1 to the PM and Arf activation	Cyth-1, -2, -3 (Cyth-4: ND)	ND	PD, IP(exo)	[78]
** *Cytoskeleton* **					
Actin cytoskeleton	・ Recruitment of cytohesin-1 phosphorylated by protein kinase C to the actin cytoskeleton during β2 integrin-mediated cell adhesion of T lymphocytes	Cyth-1 (Cyth-2, -3, -4: ND)	ND	CoS	[79]
Actinin-1	・ Neurite outgrowth in N1E-115 cells through potentiation of Arf6 in the growth cone	Cyth-2 (Cyth-1, -3, -4: ND)	PH + PB	PD, IP(exo), IP(endo)	[80]
** *Neurodegenerative disease-related* **					
SOD1	・ Neurotoxic effects through enhanced ER stress and reduced autophagic flux	Cyth-1, -2, -3 (Cyth-4: ND)	ND	IP(exog)	[33]
** *Others* **					
Aldolase	・ Actin cytoskeleton-dependent cell morphology and redistribution of acidic vesicles by forming a protein complex with cytohesin-2 and V-ATPase	Cyth-2 (Cyth-1, -3, -4: ND)	PH	PD, SPR	[81]
C1orf106	・ Maintenance of adherence junctions in intestinal epithelial cells by limiting Arf6 activation through ubiquitin-mediated degradation of cytohesin-1	Cyth-1, -2, -3 (Cyth-4: ND)	CC	MS(exo), IP(exo)	[82]

Abbreviations. *1 Cyth: Cytohesin; CC: coiled coil domain; PB: polybasic region; PH, pleckstrin homology domain; ND, not determined. *2 CoS, co-sedimentation assay; DRAP-FRET: doner recovery after acceptor photobleaching-fluorescence resonance energy transfer microscopy; IP(endo), immunoprecipitation of endogenous proteins; IP(exo), immunoprecipitation of exogenously expressed proteins; M2H, mammalian two hybrid assay; Ms(exo), mass spectrometry using immunoprecipitates of exogenously expressed proteins; MST, microscale thermophoresis; PD, pull-down assay; SPR, surface plasmon resonance binding assay; Y2H, yeast two hybrid assay.

**Table 2 ijms-23-05087-t002:** Summary of physiological roles of cytohesins in neurons.

Neuronal Processes	Functions	Cell Types	Experimental Approaches	Refs
Axon outgrowth	The cytohesin-2-Arf6 pathway negatively regulates axonal extension and branching of hippocampal neurons through downstream activation of phosphatidylinositol 4-phosphate 5-kinase α and phospholipase D.	・ Primary rat hippocampal neurons	Overexpression	[28,108]
Axon pathfinding	Cytohesin family members differentially regulate the responsiveness of commissural axons of dorsal spinal cord neurons to the repellent Slit during midline crossing: Cytohesin-2 suppresses Slit-mediated repulsion by inhibiting the surface expression of Robo before axons reach the midline, whereas cytohesin-1 and cytohesin-3 mediate Robo1 recycling to the plasma membrane and increase Slit response, allowing axons to cross and exit the midline.	・ Primary neurons from mouse dorsal spinal cord ・ Explant culture of embryonic mouse spinal cord	Knockdown	[29]
Dendrite development	The cytohesin-2-Arf6 pathway negatively regulates dendritic arborization of hippocampal neurons partly through a Rac1-dependent manner.	・ Primary rat hippocampal neurons	Overexpression	[27]
	Cytohesin-2 positively regulates dendritic extension of hippocampal neurons.	・ Primary mouse hippocampal neurons	Knockdown	[37]
	Cytohesin-1 and Arf6 participate in the extension of the apical dendrite of hippocampal pyramidal cells into the stratum lacunosum-moleculare.	・ Pyramidal cells in the mouse hippocampus	In utero electroporation Knockdown	[109]
Presynapse	The cytohesin-Arf6 pathway regulates the readily releasable pool of synaptic vesicles and recycling pathway of retrieved synaptic membrane to reform synaptic vesicles in hippocampal neurons.	・ Primary rat hippocampal neurons	SecinH3 Knockdown	[32]
	Cytohesin-1 mediates the basal synaptic transmission at the Xenopus neuromuscular junctions and Aplysia sensory-to-motor synapses.	・ Primary Xenopus spinal motor neurons ・ Primary Aplysia pedal ganglion sensory neurons	Overexpression	[30,31]
Postsynapse	Cytohesin-2 may regulate the intracellular trafficking and surface expression of group I mGluRs through the protein complex formation with group I mGluR via tamalin.	・ Primary rat hippocampal neurons	Overexpression	[41]

**Table 3 ijms-23-05087-t003:** Summary of the roles of cytohesins in neurological disorders.

Diseases	Functions	Cell types	Experimental approaches	Refs
Chronic pain	The cytohesin-2-Arf6 pathway mediates group I mGluR-dependent central sensitization at postsynapses in dorsal horn neurons of the mouse spinal cord.	Mouse spinal cord dorsal horn neurons	cKO mice SecinH3	[23]
Amyotrophic lateral sclerosis	Cytohesins promote mutant SOD1-induced neurotoxicity upstream of Arf1 and Arf5 through the interaction with mutant SOD1.	Primary rat spinal cord neurons	Overexpression Knockdown SecinH3	[33]
	Cytohesins promote mutant TDP-42-induced neurotoxicity through enhanced endoplasmic reticulum stress and reduced autophagic flux.	SH-SY5Y neuroblastoma cell line	SecinH3	[34]
Alzheimer’s disease	Cytohesins regulate FRMD4A-dependent secretion of tau dimers into the extracellular space.	Primary mouse cortical neurons HEK293 cells	SecinH3	[35]

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
