# Peer review of "Physiological and Pathological Roles of the Cytohesin Family in Neurons"

_ijms, 2022, doi:10.3390/ijms23095087_

Round 1

Reviewer 1 Report

This review provides an overview of the cytohesin proteins and their role in neuronal development and neurodegeneration. While being a very specific review (given the focus on a particular family of proteins), this review is well written and easy to follow through. The tables provide a first overview, followed by more detailed sections below.

One suggestion I have is to add a section on how cytohesin/Arf interaction modulates intracellular trafficking. Since many described processes in this review rely on intracellular trafficking (e.g. neurite growth, pathfinding, GluR-recycling and, when disrupted, neurodegeneration) this would be an overspanning theme connecting the discussed topics. Therefore I think it would be interesting to add a section on how cytohesin/Arf interaction could regulate intracellular trafficking (and a overview figure would help to give a quick summary) and briefly link it then to the following topics on neurodevelopment and neurodegeneration.

I only have one very minor comment on this well written review, which is a typo in line 368 for "GluR1/5 dependent".

Author Response

First of all, we would like to express our appreciation for valuable and constructive comments of the reviewers and editor. We have carefully and sincerely addressed all these comments and revised our manuscript as described below. Revisions made to the manuscript were marked using the Track Change function in MS word. As a result, we believe that these revisions have significantly improved our manuscript

We look forward to hearing from you that this revised manuscript will be acceptable for the publication in International Journal of Molecular Science.

Thank you very much for your consideration.

Sincerely yours,

Hiroyuki Sakagami, MD & PhD

Department of Anatomy,

Kitasato University School of Medicine

1-15-1 Kitasato, Minami-ku, Sagamihara 252-0374, Japan.

E-mail: sakagami@med.kitasato-u.ac.jp

Reviewer #1

(Comment) One suggestion I have is to add a section on how cytohesin/Arf interaction modulates intracellular trafficking. Since many described processes in this review rely on intracellular trafficking (e.g. neurite growth, pathfinding, GluR-recycling and, when disrupted, neurodegeneration) this would be an overspanning theme connecting the discussed topics. Therefore I think it would be interesting to add a section on how cytohesin/Arf interaction could regulate intracellular trafficking (and a overview figure would help to give a quick summary) and briefly link it then to the following topics on neurodevelopment and neurodegeneration.

(Response) We appreciate the reviewer’s instructive suggestion. Because most previous studies have focused primarily on the mechanisms by which the cytohesin-Arf pathways regulates actin cytoskeleton reorganization in various cellular processes associated with morphological changes, such as cell spreading, cell migration, and tubule formation, our knowledge about the mechanisms of cytohesin-Arf6-dependent intracellular trafficking is limited. Therefore, instead of the reviewer’s suggestion, we have added a new section (section 3) and table (table 1), in which we summarize and discuss the mechanisms of how neuronal functions are regulated by cytohesins through the protein-protein interactions. Although our response does not completely fulfill the reviewer’s comment, we hope that the reviewer would accept it.

(Comment) I only have one very minor comment on this well written review, which is a typo in line 368 for "GluR1/5 dependent".

(Response) We have corrected it in the revised manuscript. In addition, the revised manuscript has been checked by a professional English editing service.

Reviewer #2

(Comment 1). Line 83-84, This review concentrates on the physiological and pathological roles of the cytohesin-Arf pathway in neurons, and further discusses its potential therapeutic target for neurological disorders. I will suggest using another appropriate scientific word instead of concentrates.

(Response) In this revision, we have replaced ‘concentrates’ with ‘focus on’. If this usage were not suitable, we would appreciate if the reviewer could kindly suggest the appropriate word.

(Comment 2) Line 176, 183,191, Slit and semaphorin. Please check S capital is necessary.

(Response) Since ‘Slit’ is the formal mouse gene name, we have left the first letter of Slit capitalized n this revision.

(Comment 3) Section 5. Pathological roles of the cytohesin family in the CNS, I will suggest to authors to expand table 1 with more references related amyotrophic lateral sclerosis and Alzheimer’s disease related study.

(Response) Although we described surrounding evidence suggesting the relationship of Arf6 with molecules related to neurodegenerative diseases in normal neuronal processes in the text, experimental evidence for the functional involvement of cytohesins in these processes is still lacking. Therefore, we have not added these references to table 1 that summarizes the functional roles of cytohesins.

(Comment 4) C9ORF72 some places write normal font and some places in italic. Please write it consistently throughout of the manuscript.

(Response) According to the guideline for formatting gene and protein names, we have used italicized symbol ‘C9ORF72’ for the gene and ‘C9ORF72’ for the protein. Thus, we have kept them in this revision.

(Comment 5) Line 444, Aβ if first time use, better to use abbreviation.

(Response) In this revision, we have added the definition of the abbreviation for Ab in the place when it is first used.

(Comment 6) 5.2.2. Alzheimer’s disease author can use short form (AD) with abbreviation, then use short form throughout of the manuscript.

(Response) According to the reviewer’s suggestion, we have used AD as an abbreviation for Alzheimer’s disease.

(Comment 7) Please write your conclusion more concise according to your content. I think some information is repeated in concluding remarks section. please check, revise, and confirm.

(Response) According to the reviewer’s comment, we have deleted the sentences in the conclusion section that are redundant and overlapped with the text. As a result, we believe that the conclusion is clear and concise for readers.

(Comment 8) Before revision version submission I will suggest checking all typos and grammatical errors throughout of the manuscript.

(Response) In this revision, we have checked and corrected typos and grammatical errors as carefully as possible. Furthermore, the manuscript has been checked by a professional English proofreading and editing service. Corrections are shown in red using the Track Change function in MS word. 

Reviewer 2 Report

Dear Author,

Thanks for your review article entitled'' Physiological and pathological roles of the cytohesin family in neurons'' I read this review with great interest and felt it has well written although, it needs some modification for consider in this reputed journal. 

Comments and suggestions

  1. Line 83-84, This review concentrates on the physiological and pathological roles of the cytohesin-Arf pathway in neurons, and further discusses its potential therapeutic target for neurological disorders. I will suggest using another appropriate scientific word instead of concentrates.
  2. Line 176, 183,191, Slit and semaphorin. Please check S capital is necessary.
  3. Section 5. Pathological roles of the cytohesin family in the CNS, I will suggest to authors to expand table 1 with more references related amyotrophic lateral sclerosis and Alzheimer’s disease related study.
  4. C9ORF72 some places write normal font and some places in italic. Please write it consistently throughout of the manuscript.
  5. Line 444, Aβ if first time use, better to use abbreviation.
  6. 5.2.2. Alzheimer’s disease author can use short form (AD) with abbreviation, then use short form throughout of the manuscript.
  7. Please write your conclusion more concise according to your content. I think some information is repeated in concluding remarks section. please check, revise, and confirm.
  8. Before revision version submission I will suggest checking all typos and grammatical errors throughout of the manuscript.

Author Response

(The authors gave the same response as above.)
